# REMOVING HIGH FREQUENCY INFORMATION IMPROVES DNN BEHAVIORAL ALIGNMENT

**Max Wolff**[1]**, Evgenia Rusak**[2]**, Wieland Brendel**[1,3,4]**,**
m.wolff1621@gmail.com,
Max-Planck Institute for Intelligent Systems[1], Germany     University of Tübingen, Germany[2]
ELLIS Institute Tübingen[3]     Tübingen AI Center[4]

## ABSTRACT

Despite their increasingly impressive performance and capabilities, to date there still exists a significant misalignment between Deep Neural Networks (DNNs) and human behavior. A large body of research exists identifying misalignments and exploring where they arise from, with some work attributing it to the fact that humans and DNNs use the frequency spectrum of images differently. In this paper, we show that removing high-frequency information by applying blur and resize transformations to images before being fed to a DNN dramatically improves its alignment with humans according to shape-bias and error-consistency. Specifically, a ViT-H-14 OpenCLIP model tested on blurred images ($\sigma = 2.5$) achieves an error-consistency with humans of $\kappa = 0.37$, halving the current gap between DNN-human and human-human error-consistency. While these operations do affect a model's accuracy, we present preliminary evidence for an alignment-accuracy tradeoff, and note that moving forward, practitioners may have to choose between having a model with superhuman accuracy and one that behaves like a human.

## 1 INTRODUCTION

There is no doubt that Deep Neural Networks (DNNs) are becoming increasingly capable, and will play an large role in society in the future. At the same time, however, there is a large body of research exposing the many ways DNNs exhibit non-human-like and very often puzzling behavior. Ilyas et al. (2019); Szegedy et al. (2013) show that DNN vision models rely on "adversarial" features that humans do not use or understand. Geirhos et al. (2018) show that when presented with stimuli with conflicting shape and texture cues, humans will classify an image according to the shape cue (shape-biased), while DNNs will classify according to the texture cue (texture-bias). Geirhos et al. (2020; 2021), using the notion of *error-consistency* (Cohen, 1960), show that humans and DNNs exhibit systematically different behavior with respect to what kind of images they find difficult, again indicating that humans and DNNs likely implement very different strategies when they solve tasks. Additionally, several works approach the behavior gap between DNNs and humans from a Fourier perspective, and find that DNNs rely much more heavily on high-frequency features than humans do, which can potentially cause robustness and alignment issues (Yin et al., 2019; Subramanian et al., 2023; Li et al., 2023).

That being said, some progress has been made in closing the behavior gap between humans and DNNs. Geirhos et al. (2021); Dehghani et al. (2023) show that very large models trained on a huge, diverse dataset (CLIP (Radford et al., 2021) and ViT-22B) do exhibit a higher shape-bias and error-consistency with humans. Very recent work shows that generative diffusion models such as Imagen (Saharia et al., 2022) achieve the most human-like behavior (w.r.t. error-consistency with humans) recorded. Interestingly, they note that Imagen also displays a bias towards low-frequency features compared to other DNNs. Their hypothesis is that this could be the result of adding diffusion noise during training, or the generative objective.

We here consider a different hypothesis, namely that Imagen's $64 \times 64$ input resolution, and thus the focus on low-frequency information relative to the full $224 \times 224$ resolution, is the core reason

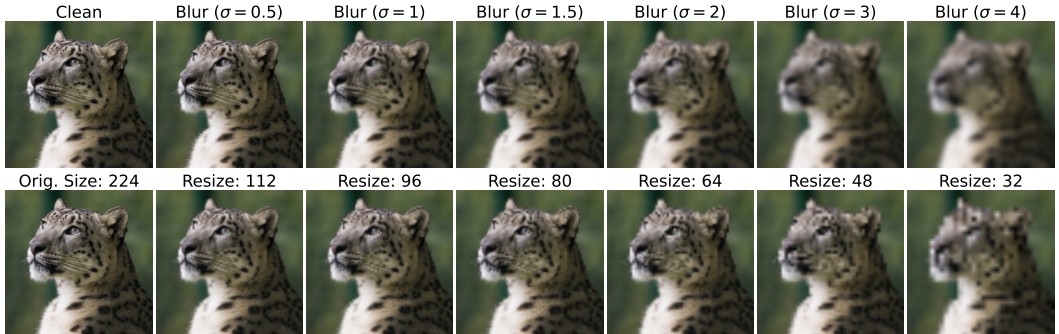

Figure 1: **Stronger blurring or resizing with a higher ratio both remove high-frequency information**. We visualize the effect of increasing the standard deviation $\sigma$ when using the Gaussian blur transformation (top row) or increasing the resize ratio (bottom row). In both cases, fine-grained image details are removed.

behind its increased human-like behavior. To this end, we test how low-pass filtering of the input of DNNs at test time affects human alignment. We find that doing so can drastically improve a DNN's alignment with humans. A ViT-H-14 OpenCLIP model tested on blurred images ($\sigma = 2.5$) achieves a Cohen's $\kappa$ of 0.37, compared to Imagen's 0.31 (Jaini et al., 2023), halving the gap between DNN-human and human-human alignment (See Table 1). Low-pass filters like blurring or resizing, however, come at the cost of accuracy: after all, removing high-frequency features from images will affect accuracy if the model has learned to rely on them. While this effect might get smaller with fine-tuning or improved models, we hypothesize that there might exist a more general principle of an alignment-accuracy trade-off, where a model achieving a superhuman accuracy on corrupted images is bound to be partially misaligned with humans, as superhuman accuracy would imply the use of non-human-like features and heuristics.

## 2 METHODS

The goal of this paper is to investigate how removing high-frequency information affects human-machine behavioral alignment. In this section, we give an overview over the datasets and models we studied, as well as our methods regarding how we remove high-frequency information.

### 2.1 ALIGNMENT METRICS

To measure behavioral consistency with humans, we use the 17 datasets from the `model-vs-human`[1] package (Geirhos et al., 2021). These datasets include clean ImageNet-1k (Russakovsky et al., 2015) images grouped into 16 ImageNet 'super-classes' (airplane, bear, bicycle, bird, boat, bottle, car, cat, char, clock, dog, keyboard, knife, oven, truck), as well as the same images after being corrupted using a variety of synthetic image distortions under various strengths. In addition, the benchmark contains stylized, edge-filtered, silhouettes, cue-conflict (see **Shape Bias** below) images and sketches. Importantly, corresponding human classification responses data for each image are provided, which allows for direct, quantitative, behavioral comparisons between humans and machines.

**Shape Bias**: We follow the experimental setup of (Geirhos et al., 2018). Their synthetic dataset contains images generated using neural style transfer (Gatys et al., 2015) where the content (shape) of one target image is combined with the texture (style) of another target image. There are 1200 images in the dataset, created from the sixteen classes listed above. Shape bias has been defined as the percentage of the time a model classifies images from the cue-conflict dataset according to shape, if the image has been classified either as the shape or the texture class.

---

[1]https://github.com/bethgelab/model-vs-human

| model | error consist. ↑ | shape bias ↑ | OOD acc. ↑ |
|---|---|---|---|
| Humans (avg) | 0.43 | 0.96 | 0.72 |
| OpenCLIP ViT-H-14 | 0.28 | 0.60 | 0.78 |
| ViT-22B-384 | 0.26 | 0.87 | 0.80 |
| Imagen | 0.31 | 0.99 | 0.71 |
| OpenCLIP ViT-H-14 Blurred ($\sigma = 2.5$) [ours] | 0.37 | 0.96 | 0.72 |
| OpenCLIP ViT-H-14 Resized (64x64) [ours] | 0.35 | 0.91 | 0.75 |

Table 1: **We observe improved error consistency and shape bias after either a blurring or a resizing transformation.** Notably, OpenCLIP ViT-H-14, when blurred using $\sigma = 2.5$, displays an error consistency of $\kappa = 0.37$, the highest which has been reported to date. At the same time, blurring and resizing transformations cause a drop in OOD accuracy of 6p.p. or 3p.p., respectively.

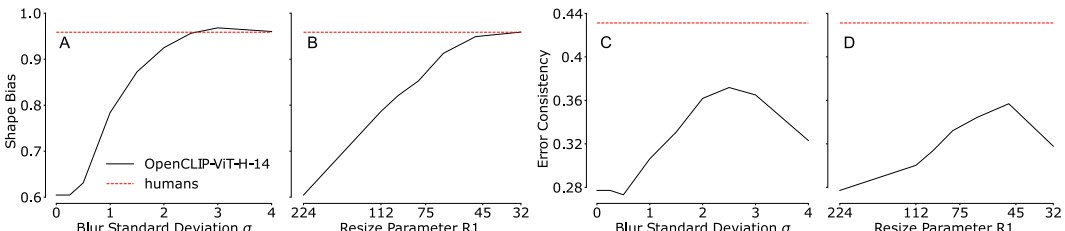

Figure 2: **Removing high-frequency information from an image improves behavioral alignment.** Gaussian blurring [A+C] and down-scaling the image to a smaller size $R1 \times R1$ with a subsequent up-scaling back to the original size [B+D] both lead to higher shape bias (A+B) and error consistency (C+D). While the shape strictly goes up when using either transformation, the error-consistency reaches a maximum 'critical point' and declines afterwards.

**Error Consistency**: This metric indicates the extent to which two decision makers (e.g., a model and a human observer) make errors on the same images (Cohen, 1960; Geirhos et al., 2020; 2021). Error consistency is measured on the data provided by the `model-vs-human` package.

**OOD Accuracy**: This metric measures the aggregate accuracy of a model or a human on the 17 datasets provided by the `model-vs-human` package.

## 2.2 MODELS AND TRANSFORMATIONS

We evaluate models trained on LAION-2B available through the open-source package OpenCLIP repository (Schuhmann et al., 2022; Radford et al., 2021; Cherti et al., 2023; Ilharco et al., 2021). In our experiments, we evaluate CLIP in the zero-shot setting using the 16 `model-vs-human` classes and the standard 80 prompt averaging scheme.

For blur and resize transformations, we use `torchvision` (Marcel & Rodriguez, 2010; Paszke et al., 2017) implementations of `GaussianBlur` and `Resize` with bi-cubic interpolation. A higher standard deviation $\sigma$ indicates a higher blurring strength. When resizing images, we perform two steps: First, we resize an image with an original size of $R_0 \times R_0$ to the desired resolution $R_1 \times R_1$. Afterwards, we resize the image back to the model's original resolution $R_0 \times R_0$. A higher 'resize strength' means resizing to a lower resolution, i.e. a larger fraction $R_0/R_1$. For example, if a model's input resolution is $224 \times 224$, resizing to $64 \times 64$ would mean a stronger resize strength than resizing to $112 \times 112$. We visualize the strengths of blur and resize transformations that we use in Fig. 1.

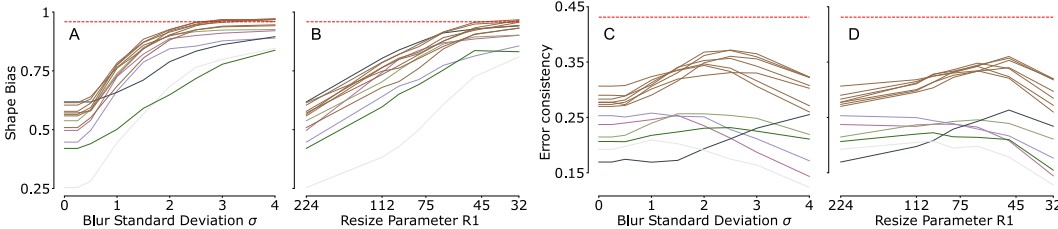

Figure 3: **Removing high-frequency information from an image improves behavioral alignment for a wide range of models.** We measure shape-bias and error-consistency for ResNet; SWSL, BiT-M, ViT, Noisy Student, and a variety of OpenCLIP models. We find that in general, shape-bias [A+B] increases with blur and resize strength, while error-consistency [C+D] usually increases at first before dropping off again.

## 3 RESULTS

### 3.1 EFFECT OF BLUR AND RESIZE STRENGTH ON A SINGLE OPENCLIP MODEL

First, we examine the role that blur and resize strength have on a model's behavioral alignment with respect to the three alignment metrics we consider. To achieve this, we first measure the error consistency and the shape bias of an OpenCLIP-H-14 model across a range of blur and resize strengths. Unsurprisingly, increasing either blur or resize strength monotonically increases shape bias (see Figure 2). Since textures are high frequency features, transformations which remove high frequency information should intuitively also bias a model towards using low frequency features, thereby increasing the model's shape bias. Curiously, neither of these transformations have a significant effect on the model's OOD accuracy: OOD accuracy dips just 6 percent points after blurring at a relatively high $\sigma = 2.5$ and only 3 percent points when resizing the image to $64 \times 64$, Table 1.

Next, we analyze the effect of the blurring and resizing transformations on the model's error consistency. Interestingly, the model's error consistency rises from $\kappa = 0.28$ to $\kappa = 0.37$ (after blurring with $\sigma = 2.5$) and $\kappa = 0.35$ (after resizing to $64 \times 64$), respectively (see Fig. 2). For reference, the previous highest error consistency has been reported for Imagen (Saharia et al., 2022) at $\kappa = 0.31$.

### 3.2 THE GENERAL EFFECT OF BLUR AND RESIZE

A natural next question to ask is whether the blur and resize transformations only increase human-model error consistency for the OpenCLIP ViT-H-14 model, or whether this is a more general phenomenon which applies to other models. To answer this question, we test several models included in the `model-vs-human` package, as well as other OpenCLIP models to see whether our results from the previous section generalize. Overall, we find that while the blurring and resizing transformations generally increase error consistency, the effect is particularly strong for the OpenCLIP models, especially those with patch sizes of 14 instead of 32 (Fig. 3). The reason might be that a patch-size of 14 biases a model towards high-frequency texture-like features which are strongly affected by these transformations.

### 3.3 THE RELATIONSHIP BETWEEN OOD ACCURACY AND ERROR CONSISTENCY

From an application point of view, ideally a model is as accurate as possible, both when classifying in- as well as out-of-distribution samples. In addition, from an alignment perspective, high error-consistency between humans and machines is also desirable, because it implies an alignment between their decision processes. In other words, in this 'ideal scenario,' a model should correctly classify as many images as possible and in the case that the model does make an error, this behavior should be similar to how a human would make an error. However, we observe a trade-off between accuracy and error consistency with humans (see Fig. 4).

What is the relationship between human-like error consistency and superhuman OOD accuracy? Machines surely have an advantage in some fine-grained classification tasks, like discriminating certain fine-grained shading differences, and they are able to utilize features not consciously avail-

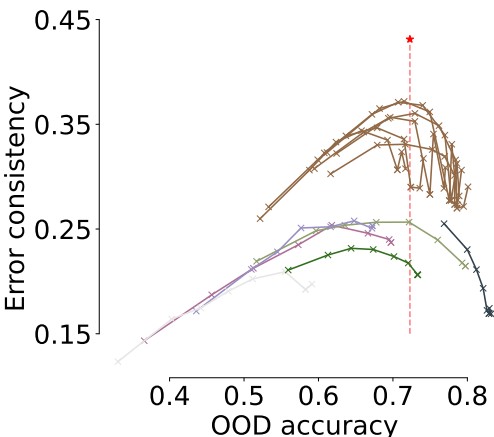

Figure 4: **The OOD accuracy and error consistency are anticorrelated for very high OOD accuracy numbers.** In this experiment, we apply a variety of blur strengths to images before passing them through the models listed in Appendix A, and plot their OOD accuracy on the transformed images versus the error consistency. For the family of OpenCLIP models (brown), models tend to achieve the highest error consistency when their OOD accuracy coincides with that of humans (dashed red line). We observe an anticorrelation between OOD accuracy and error consistency for accuracies beyond the one measured for humans.

able to the human visual system. A curious feature of the human-vs-machine benchmark is that here, the labels are mostly derived from natural images, which are then transformed through various filters. Hence, there exist images in the benchmark that humans do not classify correctly above chance. In that case, we hypothesize that there might be a fundamental trade-off between human alignment and accuracy: a machine that reaches super-human accuracy is bound to use features that humans do not use, which should generally result in a misalignment in their behavior. This is not a property of the alignment metric, which inherently corrects for observers with differential accuracy. Instead, it's the result of the features being used by DNNs. By carefully imposing limitations on the model (in our case restricting the input frequency domain), however, more human-like features may be extracted, albeit at the cost of accuracy.

## 4 CONCLUSION

In this work, we investigated how simple image transformations such as blurring and resizing, both of which remove high-frequency information from images, affect the behavioral alignment of humans and DNNs. We observed drastically improved error-consistency between humans and DNNs as well as higher shape-bias when high-frequency information was removed. We also found a decrease in OOD accuracy, which we hypothesise to hint to a more fundamental trade-off between human alignment and performance. We believe this has two implications: (1) from an alignment perspective, a model may be only as generally 'useful' as it is 'useful' on low-frequency information; (2) we predict that practitioners might soon have to make a choice between alignment and accuracy, as a superhuman accuracy is likely impossible without using non-human-like heuristics.

In the future, we look forward to: conducting a more systematic study across different model types and pre-training schemes; doing a more in-depth analysis on the relationship between a model's Fourier spectrum and alignment with humans; performing our blur and resize experiments w.r.t. a larger suite of alignment metrics; and exploring the accuracy-alignment relationship more fully, including improving the trade-off through fine-tuning.

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

## A  APPENDIX

We present an overview of the models we used in this work in Table 2.

| model | architecture |
|---|---|
| ResNet (He et al., 2016) | ResNet101 |
| SWSL (Yalniz et al., 2019) | ResNeXt-101 |
| BiT-M (Kolesnikov et al., 2020) | ResNet-101x1 |
| ViT (Dosovitskiy et al., 2020) | ViT-L-16 (IN1K and IN21K) |
| Noisy Student (Xie et al., 2020) | EfficientNet-L2 |
| OpenCLIP (Ilharco et al., 2021) | ViT-B-32 |
|  | ViT-B-16 |
|  | ViT-L-14 |
|  | ViT-H-14 |
|  | ViT-g-14 |
|  | ViT-G-14 |
|  | ConvNext-L |

Table 2: Architectures and the color coding used for each model type. We adopt these from Geirhos et al. (2021).

