# OpenReview forum: "Removing High Frequency Information Improves DNN Behavioral Alignment"
_ICLR.cc/2024/Workshop/Re-Align — ICLR 2024 Workshop Re-Align Poster_

### Official Review · Reviewer_atKD · 2024-02-22
**Interesting observation that clearly improves behavioural alignment of humans and DNNs**

**Rating:** 3
**Fit:** 3
**Confidence:** 3

**Workshop Review:**

This short paper describes a simple transform to improve alignment of DNNs to human classification behaviour. The authors simply remove high spatial frequency information from the images at test time by blurring the image or resizing to a small image size and back. This simple transformation improves shape bias and error consistency of DNNs substantially, especially of OpenCLIP trained networks. These are models that are already better aligned with human behaviour.

In general, I believe this paper will be of strong interest for the Realign workshop and makes a genuinely interesting observation and the tests are done for full complexity, SOTA models. Blurring the image is not a new idea for this purpose, but this working so well now for the OpenCLIP models did surprise me.

There are a few details that I believe would be good to discuss at the workshop:
- For the OOD accuracy results: Some of the image distortions are also about filtering the images. Thus, splitting up the results into the original distortions would be informative.
- Some comparison of the blur strength to human visual processing would be interesting. For example, a conversion into degrees of visual angle in Geirhos et al’s experiment would be great to have for reference. Or perhaps how this compares to peripheral loss of resolution or optical blur in humans.
- The error-consistency tries to be invariant to the accuracy of the two compared classifiers, but this is not perfect. Additionally, the metric of course does not contain all the information. To understand the findings in more detail, I would love a comment on whether additional human-like errors are made or existing errors now shift to the category that humans chose.

But these points are about how to interpret and use these results and do not speak against having the findings presented at the workshop.

**Reason For Not Giving Higher Score:**

N/A

**Reason For Not Giving Lower Score:**

I think the observation is interesting and improves the alignment of humans and DNNs and I did not see any reason to not believe the essential results.

**Reviewer Domain:**

cognitive science

---

### Official Review · Reviewer_5MZm · 2024-02-22
**Interesting idea but lacking clarity**

**Rating:** 2
**Fit:** 3
**Confidence:** 2

**Workshop Review:**

This paper analyzes a simple and interesting idea, which is that images blurred in some manner will be classified by DNNs in a manner more similar to humans than the unaltered images. This is due to the known phenomenon of neural networks exploiting high frequency spurious features in classifying images, while humans do not perceive or use these features. The finding in this paper is a neat one - I am not sure if the study is quite comprehensive enough but I see the value of accepting.

Clarity: Some clarity is lacking throughout the paper:
1. In Table 1, what two models is error consistency measured between? In particular, what does error consistency of humans mean?
2. The legend for Figure 4 is included in the appendix although there could have been space to include it in the main paper. This hinders the clarity of this result.

Correctness: The results seem correct to me.

Novelty: The novelty is somewhat minimal but I think this observation sets the stage for further work to simultaneously improve OOD accuracy and error consistency.

Interest to the community: As stated in novelty, I would hope that this work generates further research into pushing the pareto frontier of this tradeoff.

**Reason For Not Giving Higher Score:**

The analysis is quite limited, clarity is lacking.

**Reason For Not Giving Lower Score:**

I think the finding is a neat, if not surprising, result and it merits sharing at the venue.

**Reviewer Domain:**

machine learning

---

### Official Review · Reviewer_JB5X · 2024-02-23
**Simple but clear hypothesis and convincing analysis.**

**Rating:** 3
**Fit:** 3
**Confidence:** 2

**Workshop Review:**

**Summary**

This paper hypothesizes that the high shape bias achieved by Imagen, as shown by a recent paper, is not because of the better alignment of generative objectives of Gaussian noise as an augmentation, but instead because of the 64x64 image size. To test this hypothesis, they measure shape bias and OOD error consistency of VITH-OpenCLIP models trained with a smaller equivalent image size and blurred images, and find that shape bias increases with amount of blur and resize, while OOD accuracy increases till a point and then drops. Their results suggest that for models that achieve superhuman accuracy on OOD datasets, there is a tradeoff between error consistency and alignment i.e. usage of low frequency information and shape bias.

**Clarity**
The paper is very clear and easy to follow.

**Correctness**
All methods and analysis are correct to the best of my knowledge.

**Novelty and interest to the community**
The observed relationship between blurriness of an image at test-time and error consistency is novel and of interest to the community.

**Reason For Not Giving Higher Score:**

NA

**Reason For Not Giving Lower Score:**

The paper presents a clear hypothesis and tests that hypothesis convincingly with their analysis.

**Reviewer Domain:**

machine learning

---

### Decision · Program_Chairs · 2024-03-02

Accept (Poster)